# Occupational radiation exposure dose and associated factors among radiology personnel in Eastern Amhara, Ethiopia

**Tadesse Guangul Alemayehu**[1], **Getahun Gebre Bogale**[2], **Getaw Walle Bazie**[3]*

**1** Dessie Health Science College, Dessie, Ethiopia, **2** Department of Health Informatics, College of Medicine and Health Sciences, School of Public Health, Wollo University, Dessie, Ethiopia, **3** Department of Epidemiology and Biostatistics, College of Medicine and Health Sciences School of Public Health, Wollo University, Dessie, Ethiopia

* getaw4jesus@gmail.com

**Data Availability Statement:** All relevant data are within the manuscript and its Supporting Information files.

## Abstract

### Background

Ionizing radiation is being used more frequently in medicine, which has been linked to recognized biological effects such as cancer and mortality. Radiology services are becoming more widely available in Ethiopian health facilities but there is no compiled record of worker's radiation dose. So, assessing the magnitude and identifying the associated factors of occupational radiation exposure dose among radiology personnel help to design strategies for radiation protection.

### Objective

The study was designed to assess the occupational radiation exposure dose and associated factors among radiology personnel in eastern Amhara, northeast Ethiopia, 2021.

### Methods

Cross-sectional study was conducted from March 25 to April 30, 2021, in 57 health institutions among 198 radiology personnel. The study comprised all eligible radiology personnel. The data were collected using an electronic-based (Google form) self-administered questionnaire, and document review. The data were entered into an excel spread sheet and then, exported to Stata 14 software. Linear regression model was used to analyse the data after checking its assumptions. Variables with a p-value < 0.25 were entered into a multiple linear regression analysis, and those with a p-value < 0.05 were judged significant. VIF was used to check for multi-collinearity. Coefficient of determination was used to check the model fitness.

### Results

The mean (± SD) annual shallow and deep dose equivalents of radiology personnel were 1.20 (± 0.75) and 1.02 (± 0.70) mSv, respectively. Body mass index (β = 0.104, 95% CI: 0.07, 0.14), practice of timing (β = -0.43, 95% CI: -0.73, -0.13), working experience (β =

**Funding:** The authors received no specific funding for this work.

**Competing interests:** The authors have declared that no competing interests exist.

**Abbreviations:** BMI, Body Mass Index; CT, Computed Tomography; ERPA, Ethiopian Radiation Protection Authority; IAEA, International Atomic Energy Agency; ICC, Intra Class Correlation; ICRP, International Commission on Radiation Protection; LMM, Linear Mixed Effect Model; MRI, Magnetic Resonance Imaging; MRT, Medical Radiology Technologist; mSv, milli-Sieverts; RPE, Radiation Protection Equipment; SD, Standard Deviation; Sv, Sievert; TLD, Thermo-Luminescent Dosimeter; UAE, United Arab Emirates; UNSCEAR, United Nations Scientific Committee on the Effect of Atomic Radiation.

-0.04, 95% CI: -0.048, -0.032), and practice of distancing (β = -0.26, 95% CI: -0.49, -0.17) were found to be statistically significant factors of annual deep dose equivalent. In addition, body mass index (β = 0.113, 95% CI: 0.08, 0.15), practice of timing (β = -0.62 95% CI: -0.93, -0.31) and, working experience (β = -0.044, 95% CI: -0.053, -0.036 had statistically significant associations with annual shallow dose equivalent.

## Conclusion

The annual dose equivalents were two times higher than the global average of annual per caput effective dose due to medical exposure. Body mass index, practice of timing, working experience, and practice of distancing were factors of occupational radiation exposure dose. Strategies focusing on increasing the skill, experience, and lifestyle of radiology personnel would be supreme important means to reduce occupational radiation exposure dose.

## Introduction

Radiation is energy that is emitted in the form of waves or particles and transferred via a medium or space. It can come from both natural and manmade sources [1–3]. Ionizing and non-ionizing radiation are the two types of radiation. Ionizing radiation is defined as radiation with the highest energy and penetration power, such as x-rays [4]. Non-ionizing radiation, on the other hand, is a form of radiation with less energy than ionizing radiation. It is used in ultrasound that applies high frequency sound waves. It is also used in magnetic resonance imaging (MRI) scanners that uses radiofrequency and does not induce ionization [5]. Unless otherwise stated, the term radiation in this document refers to ionizing radiation

According to the International Commission on Radiological Protection's (ICRP) recommendations, occupational radiation exposure can be defined using effective dose and equivalent dose [6, 7]. Workers' annual effective dose limits should not exceed 20 millisieverts (mSv) averaged over five years (100 mSv in five years), with a stipulation that no single year's exposure should exceed 50 mSv [6]. According to the Ethiopian national radiation protection authority (ERPA), the yearly workers' effective dose limit is 12 mSv per year [8]. It is preferable not to exceed the recommended yearly fetal dosage limit of 1 mSv per year for female workers [6].

The average annual radiation exposure dose from all sources for the global population is roughly 3 mSv per person per year, with medical use of radiation accounting for 19.7% (0.6 mSv) of the annual dose from all sources [9]. Medical uses of radiation account for more than 99.9% of all man-made radiation exposure, with billions of procedures performed each year in all parts of the world [10]. According to recent estimates from the United Nations Scientific Committee on the Effects of Atomic Radiation (UNSCEAR), there are four billion x-ray examinations performed worldwide each year [11].

Because of the widespread nature and volume of medical procedures, medical uses of radiation result in the largest number of occupationally exposed workers and represent the largest occupational collective dose, even though the exact number of occupationally exposed workers as a result of medical uses globally is unknown [11, 12]. Approximately 7.5 million workers are exposed to medical usage of ionizing radiations, according to a UNSCEAR 2008 report [13]. UNSCEAR estimates that over 2.5 million workers are being tracked with a total exposure of 850 person-Sv in medicine which is higher than the dose in industry and military [14].

In comparison to developed countries, the number of diagnostic and therapeutic x-ray procedures in low-income countries, such as Sub-Saharan Africa, is limited. However, for a variety of reasons, the amount of occupational radiation exposure dosage is higher in developing countries. Perhaps the most important reasons are a lack of skill and experience, a lack of radiation protection education and training, a lack of radiation protective barriers, poor machine quality (the machines are either old, conventional machines or donated used machines), a lack of equipment calibration, and a lack of machine spare parts [10, 11, 15, 16].

Radiation has been shown to have harmful biological and clinical effects, including cancer, in x-ray physicians. Radiation exposure can create biological effects that are damaging to the person who is exposed, and these effects can be both short-term (skin burn and cataract) and long-term (cancer and death) [3, 17–21].

The number of studies on the effects of radiation on radiology staff in African countries is limited. Radiation safety is a big concern in African countries for a variety of reasons. The medical use of radiation is rising, particularly in Ethiopia, yet the number of certified radiology personnel, radiation protective equipment, and radiation safety training programs are insufficient. International efforts, such as the International Atomic Energy Agency's (IAEA) training courses on radiation protection in medicine and ERPA's radiation protection guidelines, have attempted to address this gap. However, reaching all radiology personnel has proven extremely challenging [10, 15–17, 22].

Studies on occupational radiation exposure dosage and associated aspects of occupational radiation exposure dose have been undertaken on their own. However, research on the magnitude and associated factors has been sparse in Ethiopia and around the world. Even occupational radiation exposure dosage studies are mainly concerned with deep dose equivalents. Personal factors like; utilization of radiation protection equipment and health facility factors like; type of health facility and level of health facility are not undertaken so far. Studies on occupational radiation exposure and associated factors are scarce, particularly in Ethiopia. Furthermore, there has been no research on occupational radiation exposure dose in eastern Amhara. Thus, the purpose of this study was to assess the occupational radiation dose (both deep and shallow dose equivalent) and associated factors among radiology personnel in eastern Amhara.

## Methods

### Study area and period

The study was conducted in eastern Amhara, northeast Ethiopia. Eastern Amhara is found in northeast Ethiopia and composed of five zones. Overall, in the eastern Amhara, there are above 30 governmental, 40 private, and 1 non-governmental health institutions which provide diagnostic radiology services. There are above 245 radiology personnel who are currently working in eastern Amhara. The study was conducted from March 25 to April 30, 2021.

### Study design

Institutional based cross-sectional study design was employed among radiology personnel working in eastern Amhara.

### Population

The source population was all radiology personnel working in eastern Amhara private, NGO (Non-Governmental Organization), and governmental health institutions. The study population was all radiology personnel working in functional diagnostic x-ray health institutions in eastern Amhara who had exposure dose report from January 1, 2020 to December 30, 2020.

### Inclusion criteria

All radiology personnel who had radiation exposure dose report from ERPA and DosiMed testing laboratory in year from January 1, 2020 to December 30, 2020 G.C were included in the study.

### Exclusion criteria

All radiology personnel who were new employers under a year were excluded from the study.

### Sample size determination

Sample size was calculated by using single population mean formula with the assumption of 95% confidence, marginal error and variance obtained from the previous study [23].

$$n = \frac{(z\alpha/2)2*\delta2}{d2} = \frac{(1.96)2*(35)2}{(5)2} = 188$$

The final sample size including 10% non-response rate was:

n = 188+188*10/100

n = 207

Therefore, the final sample size was 207.

Where; n = required sample size, d = marginal error, z = test statistics, $\alpha$ = level of significance, and $\delta^2$ = variance.

### Sampling procedures

In the ERPA database, there were 245 registered radiology personnel currently working in eastern Amhara health institutions. Twenty-seven of them did not have an occupational radiation dose record from January 1, 2020 to December 30, 2020 and 20 of them had less than a year of experience, and 198 met the inclusion criteria. All 198 eligible radiology personnel were included in the study, although the projected sample size was 207. A list of all 57 functional diagnostic x-ray health facilities with quality control records of machines was also collected from the ERPA data source and was included in the study for the assessment of machine quality control.

### Operational definitions

- **Occupational radiation exposure dose:** the dose of radiation received by radiology personnel at the workplace during operations which was extracted and calculated from the ERPA and DosiMed testing laboratory data bases. The calculated mean (± SD) annual TLD readings in deep dose equivalent, shallow dose equivalent, and the calculated sum in collective dose were considered as occupational radiation exposure dose [12]. The dose report of radiology personnel was reported every two months.

- **Deep dose equivalent:** the dose equivalent at the depth of 10 mm for strongly penetrating radiations (high energy x-rays)

- **Shallow dose equivalent:** the dose equivalent at the depth of 0.07 mm for weakly penetrating radiations (low energy x-rays)

- **Collective dose:** the sum of all individual effective doses over the time period due to ionizing radiation. The individual effective dose was calculated from the deep dose equivalent (Hp10) and shallow dose equivalent (Hp 0.07), which indicates the personal dose at a tissue depth of

10 mm and 0.07 mm. Using the lead apron, does not affect the individual effective dose as the TLD is put under the apron.

- **Radiology personnel:** the common name given to health professionals working in medical radiology unit with different level of education. This includes; darkroom workers (pre-college and pre-university), radiographers (diploma), medical radiology technologist (B.Sc.), and radiologist (specialist) [24].

- **Quality control test:** refers to the specific test required to ensure effective and safe equipment performance [25].

- **Level of health facility:** according to this study level of health facilities means the hierarchy of health facilities from medium clinic to referral hospitals that provides radiology services.

## Data collection tools

The data was collected by using self-administered questionnaire and document review. The questionnaire was comprised from socio-demographic and economic, personal, occupational, institutional and environmental factors based on ERPA guidelines, ICRP recommendations, and different literatures [10, 15, 16, 22, 24] to address all important variables which affect occupational radiation exposure dose. The data of the respondents was collected utilizing a Google form, which is an electronic-based data collection approach.

## Data collection procedures

Document review was used to extract the annual Thermoluminescent Dosimeter (TLD) readings of all monitored radiology personnel for the year 2020 from the ERPA and DosiMed testing laboratories' databases. In addition, utilizing document review, machine quality control tests were extracted from the ERPA database. The annual TLD readings were taken from the ERPA and DosiMed testing laboratory (secondary data) for this study. Computerized Vinten Solaro machine was used by the two organizations as a TLD reader. The quality assurance of each TLD machine was done regularly by the experts in the two organizations. Furthermore, each radiology personnel were equipped with two TLDs (when one is sent for reading, the other is used). The radiology personnel were responsible to declare the TLD they use every two months by the Authority. A pre-tested self-administered questionnaire was used to data from the respondents. The self-administered questionnaire was created in Google Forms, and the link was delivered to responders electronically, with supervisors and the principal investigator monitoring the process.

## Data quality control

A medical radiology technologist from Addis Ababa University was hired as a secondary data collector, and five radiography lecturers from Dessie Health Science College were hired as supervisors for electronic data collecting. Prior to the data collecting date, a two-day training was given for a data collector and supervisors on the objective of study, data collection methods, data collection tools, including the electronic-based data collection technique, respondents' approach, data confidentiality, and respondent's right. In addition, respondents were made aware of the data collection process. Completeness of the questionnaires was checked every day by the supervisors and principal investigator.

Pretesting of the questionnaire was undertaken among 10 radiology personnel working in Addis Ababa health institutions prior to the actual data collection to prevent information

leaking, and then necessary modifications were made based on the pretesting findings. The reliability (internal consistency) of the data collection tool was evaluated using Cronbach's alpha. The Cronbach's alpha value of the items selected in the tool was 0.8.

## Data processing and data analysis

Data from ERPA's and DosiMed testing laboratory occupational radiation dose reports and data from ERPA's machines quality control test were checked, edited and cleaned for completeness and consistency of data manually before entered in to the computer. The respondent's data from Google form was exported to Excel. Then, data from the document review as well as the data from the Google form (in Excel) were compiled into an excel database. Finally, for processing and analysis, the data were exported to Stata version 14.0 software.

Descriptive and summary statistics were used to describe data quantitatively. The ICC was calculated to fit a linear mixed effect model (LMM). However, the ICC value was 2.6%, which was too small to conduct LMM. Thus, linear regression model was used to estimate the effects of independent variables on occupational radiation exposure dose. The assumptions of linear regression were tested using scatter plots, kernel density, residual versus fitted value (rvfplot) and finally Kolmogrov-smirnov test was used to check normality objectively. The p-value of kolmogrov-smirnov test was 0.138 which shows the data were normally distributed.

In simple liner regression variable analysis, variables having p value $<0.25$ were entered to multiple analysis of linear regression analysis and in multiple linear regression analysis, variables having p value $<0.05$ were declared as significant variables for dependent variable. Variance inflation factor (VIF) was used to assess for multi-collinearity, and individual variables VIF $> 5$ and average VIF $> 10$ were considered as multi-collinearity and were eliminated from multiple linear regression analysis. Coefficient of determination ($R^2$) was used to check the model fitness. The value of $R^2$ was 0.61 which showed that the model was fitted.

## Ethical considerations

Ethical clearance was obtained from the Ethics Review Committee of College of Medicine and Health Sciences, Wollo University, Ethiopia. Then, permission and support letter were written to ERPA by Wollo University, College of Medicine and Health Science research, community service and post graduate studies co-ordinating office.

Written consent was obtained from radiology personnel and prior to enrolling any of the eligible study participants and the purpose, the benefits, and the confidential nature of the study were described for each participant. Confidentiality was assured by omitting the names of the participants. Only those that consented and proved their willingness to take part in the study were enrolled and respondent's right to withdraw from the study at any time without any penalty was assured. Confidentiality of data was maintained anonymously.

## Results

A total of 198 radiology personnel who had occupational radiation exposure dose report in the year 2020 were interviewed in this study.

## Socio-demographic and economic characteristics of respondents

From the total of 198 radiology personnel, 123 (62.12%) of them were males, 103 (52.02%) of them were single, 140 (70.71%) of them were Orthodox Christians, and 108 (54.55%) of them were radiographers. The respondent's age ranged from 22 to 70 years old, with a mean (±SD) of 33 (±11) years old (**Table 1**).

**Table 1. Socio-demographic characteristics of radiology personnel in eastern Amhara, northeast Ethiopia, 2021.**

| Variables | Categories | Frequency | Percent |
|---|---|---|---|
| **Sex** | Male | 123 | 62.12% |
| | Female | 75 | 37.88% |
| **Religion** | Orthodox | 140 | 70.71% |
| | Muslim | 58 | 29.29% |
| **Marital status** | Single | 103 | 52.02% |
| | Married | 95 | 47.98% |
| **Level of education** | Darkroom worker | 40 | 20.20% |
| | Radiographer | 108 | 54.55% |
| | MRT | 49 | 24.75% |
| | Radiologist | 1 | 0.51% |

MRT = Medical Radiology Technologist

### Practice of radiation protection

This study showed that among 198 radiology personnel, 111 (56.06%), 115 (58.08%), 119 (60.10%), and 115 (58.08%) did not wear lead apron, eye goggle, thyroid collar, and lead glove, respectively and 130 (65.66%) of them had never taken radiation protection on-job training (**Table 2**).

### Health facility characteristics

In eastern Amhara, there were 57 health institutions which provided radiology services and all of them were included in the study. From these, 33 (57.89%) of them were private health institutions. From the total of 57 health institutions, there were 56 x-rays [35(61.40%)] of them were conventional], 2 CT scan and 2 fluoroscopy machines. The machine's age ranged from 3 to 17 years old with a mean (±SD) of 10 (±4) years old.

### Quality control test results

Radiation safety assessment reports of 57 health institutions were collected from ERPA. The quality control test results of all x-ray machines were deviated in acceptable range. The KVP and timers of machines were deviated from– 5.9% to 5.61% and from -3.6% to 3.6% in average, respectively (**Table 3**).

### Occupational radiation exposure dose

The mean (± SD) annual shallow and deep dose equivalents of 198 radiology personnel were 1.20 (± 0.75) 95% CI (1.09, 1.30) and 1.02 (± 0.70) mSv 95% CI (0.92, 1.12), respectively. The annual collective shallow and deep dose equivalents were 238 person-Sv and 202 person-Sv, respectively. The females mean shallow and deep dose equivalents were 1.24 mSv and 1.05 mSv, respectively.

The mean annual shallow and deep dose equivalents of darkroom workers were 1.54 mSv and 1.32 mSv, respectively. The mean annual shallow and deep dose equivalents in private health institutions were 1.44 mSv, and 1.23 mSv, respectively (**Table 4**).

### Factors associated with occupational radiation exposure dose

In this study, a $kg/m^2$ increase in body mass index, increases the mean annual shallow and deep dose equivalents by 11.3% (95% CI: 0.08, 0.15) and 10.4% (95% CI: 0.07, 0.14),

**Table 2. Radiology personnel's utilization of radiation protective equipment by education level in eastern Amhara, northeast Ethiopia, in 2021.**

| Variable name | Label of education | | | | Total(n = 198) |
|---|---|---|---|---|---|
| Utilization of RPE* | Darkroom worker(n = 40) | Radiographer (n = 108) | MRT (n = 49) | Radiologist (n = 1) | |
| **Lead apron** | | | | | |
| Yes | 6(3.03%) | 41(20.71%) | 39(19.70%) | 1(0.51%) | 87(43.94%) |
| No | 34(17.17%) | 67(33.84%) | 10(5.05%) | 0(0.00) | 111(56.06%) |
| **Gonad shield** | | | | | |
| Yes | 9(4.55%) | 80(40.40%) | 43(21.72%) | 1(0.51%) | 133(67.17%) |
| No | 31(15.66%) | 28(14.14%) | 6(3.03%) | 0(0.00) | 65(32.83%) |
| **Eye goggle** | | | | | |
| Yes | 7(3.54%) | 37(18.69%) | 38(19.19%) | 1(0.51%) | 83(41.92%) |
| No | 33(16.67%) | 71(35.86%) | 11(5.56%) | 0(0.00) | 115(58.08%) |
| **Thyroid collar** | | | | | |
| Yes | 7(3.54%) | 35(17.68%) | 36(18.18%) | 1(0.51%) | 79(39.90%) |
| No | 33(16.67%) | 73(36.87%) | 13(6.57%) | 0(0.00) | 119(60.10%) |
| **Lead glove** | | | | | |
| Yes | 7(3.54%) | 37(18.69%) | 38(19.19%) | 1(0.54) | 83(41.92%) |
| No | 33(16.67%) | 71(35.86%) | 11(5.56%) | 0(0.00) | 115(58.08%) |
| **Practice of distancing** | | | | | |
| Yes | 31(15.66%) | 92(46.46) | 49(24.75%) | 1(0.51%) | 173(87.37%) |
| No | 9(4.55%) | 16(8.08%) | 0(0.00) | 0(0.00) | 25(12.63%) |
| **Practice of timing** | | | | | |
| Yes | 34(17.147%) | 100(50.51%) | 49(24.75%) | 1(0.51%) | 184(92.93%) |
| No | 6(3.03%) | 8(4.04%) | 0(0.00) | 0(0.00) | 14(7.07%) |
| **Radiation protection on-the-job training** | | | | | |
| Yes | 6(3.03%) | 26(13.13%) | 36(18.18%) | 0(0.00) | 68(34.34%) |
| No | 34(17.17%) | 82(41.41%) | 13(6.57%) | 1(0.51%) | 130(65.66%) |

\* RPE = Radiation Protective Equipment; MRT = Medical Radiology Technologist

respectively. On the contrary, a year increase in working experience of radiology personnel, decreases the mean annual shallow and deep dose equivalents by 4.4% (95% CI: -0.053, -0.036) and 4% (95% CI: -0.048, -0.032), respectively.

Radiology personnel's practice of timing was also statistically associated with occupational radiation exposure dose. A second increase in radiology personnel's practice of timing, decreases the mean annual shallow and deep dose equivalents by 62% (95% CI: -0.93, -0.31) and 43% (95% CI: -0.73, -0.13), respectively.

Radiology personnel's practice of distancing had significant statistical association with mean annual shallow dose equivalent of occupational radiation exposure. A centimetre increase in radiology personnel's position from the x-ray tube, decreases mean annual shallow dose equivalent by 26% (95% CI: -0.49, -0.17) (**Table 5**).

## Discussion

The TLD readings of 198 radiology personnel in eastern Amhara, north east of Ethiopia from January 1, 2020 to December 30, 2020 G.C showed that the mean (± SD) annual shallow and deep dose equivalents were 1.20 (± 0.75) mSv and 1.02 (± 0.70) mSv, respectively. The annual collective shallow and deep dose equivalents were 238 person-Sv and 202 person-Sv, respectively.

**Table 3. Machine's quality control test results of health institutions in eastern Amhara, north east of Ethiopia, 2021.**

| Variable | Percent deviation | | acceptability | |
|---|---|---|---|---|
| | Minimum (%) | Maximum (%) | Yes | no |
| **Voltage accuracy** | -5.9 | 5.61 | ✓ | |
| **Timer accuracy** | -3.6 | 3.6 | ✓ | |
| **Half value layer** | 2.4 | 3.9 | ✓ | |
| **Leakage measurement** | | | | |
| Leakage at the collimator | 0 | 0.24 | ✓ | |
| Leakage at the tube | 0 | 0.24 | ✓ | |
| **Output consistency** | 0 | 3.67 | ✓ | |
| **Collimator and beam alignment** | | | | |
| Long axis variance | -1.25 | 1.5 | ✓ | |
| Short axis variance | -1.25 | 1.25 | ✓ | |
| **Work area dose result** | Minimum(mSv) | Maximum(mSv) | ✓ | |
| Main door to x-ray department | 0.19 | 0.32 | ✓ | |
| Lead glass window | 0.17 | 0.74 | ✓ | |
| Operator's position | 0.15 | 0.68 | ✓ | |
| Door to the toilet | 0.09 | 0.28 | ✓ | |
| Door to the dressing room | 0.14 | 0.31 | ✓ | |
| Opened hatch box form the darkroom | 0.13 | 0.39 | ✓ | |
| Darkroom | 0.17 | 0.26 | ✓ | |
| Average work area dose | 0.2 | 0.4 | ✓ | |

Both the annual deep and shallow dose equivalents were below the ERPA and ICRP, 2007 recommended dose limits [6, 26]. However, the doses were two times higher than the world's population annual effective dose (0.6 mSv) due to medical use of radiation [9]. The occupational radiation exposure doses of females were higher than the recommended fetal dose level (1 mSv) [6]. The collective annual shallow and deep dose equivalents were 28% and 24% of the total medical exposure of the world population (850 person-Sv) [14] which shows the collective occupational radiation exposure doses were high. The possible reasons might be related to the type of x-ray machines, government's concern to radiation, the workload, radiology personnel's awareness and adherence to radiation protection in Ethiopia.

The mean annual deep dose equivalent was in line with the study done in Kuwait (1.03 mSv) [27] and Ghana (1.05 mSv) [28]. It was lower than the studies done in Kenya (2.15 mSv)

**Table 4. Occupational radiation exposure dose of radiology personnel in eastern Amhara, north east of Ethiopia, 2021.**

| Variables | Categories | Mean annual shallow dose equivalent (mSv) | Mean annual deep dose equivalent (mSv) |
|---|---|---|---|
| **Sex** | Male | 1.17 | 1.00 |
| | Female | 1.24 | 1.05 |
| **Education level** | Darkroom workers | 1.54 | 1.32 |
| | Radiographers | 1.31 | 1.12 |
| | MRT* | 0.70 | 0.57 |
| | Radiologist | 0.15 | 0.09 |
| **Types of health facility** | Government | 0.98 | 0.84 |
| | Private | 1.44 | 1.23 |
| | NGO* | 0.64 | 0.49 |

MRT* = Medical Radiology Technologist NGO* = Non-Governmental Organisation

**Table 5. Factors of occupational radiation exposure dose in eastern Amhara, northeast of Ethiopia 2021.**

| Mean annual deep dose equivalent | | | | |
|---|---|---|---|---|
| **Variables** | β* | **Std**. Err. | p-value | 95% CI |
| **BMI** | 0.104 | 0.018 | <0.001* | 0.07 0.14 |
| **Practice of distancing** | -0.255 | 0.121 | 0.036* | -0.49–0.017 |
| **Practice of timing** | -0.43 | 0.15 | 0.005* | -0.073–0.13 |
| **Working experience** | -0.04 | 0.004 | <0.001* | -0.048–0.032 |
| **Constant** | 0.524 | 0.519 | 0.315 | -0.500 1.547 |
| Mean annual shallow dose equivalent | | | | |
| **BMI** | 0.113 | 0.019 | <0.001* | 0.08 0.15 |
| **Practice of distancing** | -0.173 | 0.126 | 0.170* | -0.421 0.075 |
| **Practice of timing** | -0.620 | 0.158 | <0.001* | -0.93–0.31 |
| **Working experience** | -0.044 | 0.0042 | <0.001* | -0.053–0.036 |
| **Constant** | 0.760 | 0.540 | 0.161 | -0.304 1.823 |

β* = un standardized β coefficient

*: p<0.05

**: 0.05<p<0.25 for simple linear regression; BMI = Body Mass Index

[29], and Malawi (2.964 mSv) [30] (**Table 6**). The possible reason might be, in Ethiopia there is a regular safety assessment and quality control tests of all machines by the national radiation protection authority of Ethiopia. This would decrease the occupational radiation exposure dose of radiology personnel.

It was 3 times higher than the studies done in China (0.35 mSv) [31], and Ireland (0.32 mSv) [32], 2.5 times higher than the study done in Saudi Arabia (0.41 mSv) [23], 1.2 to 2.6 times higher than the studies done in UAE (0.39–0.83 mSv) [33], and 1.3 to 3.5 times higher than the studies in done in Pakistan (0.29–0.80 mSv) [34]. The possible reasons for low annual deep dose equivalent in these countries might be due to: good practice of radiation protection, adequate number of skilled radiology personnel, the quality of x-ray machines, and the availability of machine spare parts, the availability as well as the quality of radiation protection barriers, education and training.

The dose in this study was 2.5 times higher than the study done in Ethiopia on radiation exposure of diagnostic x-rays on patients and personnel in Addis Ababa by Alemu Tesfahun

**Table 6. The mean annual deep dose equivalent of selected countries in the world, 2023.**

| Countries | Radiation dose in mSv | remark |
|---|---|---|
| Ghana | 1.05 | |
| Kuwait | 1.03 | |
| Kenya | 2.15 | |
| Malawi | 2.964 | |
| China | 0.35 | |
| Ireland | 0.32 | |
| Saudi Arabia | 0.41 | |
| UAE | 0.39–0.83 | |
| Pakistan | 0.29–0.80 | |
| Ethiopia | 0.41 | Previous study |
| Ethiopia | 1.2 (Shallow dose equivalent) and 1.02 (Deep dose equivalent) | Current study |

UAE = United Arab Emirates

(0.41 mSv) [22]. The low dosage for Alemu Tesfahun's study might be due to improper utilization of thermo-luminescence dosimetry, the availability of radiation protection equipment, low workload, and ERPA's proximate control of health facilities because the main office is located there.

The study also identified factors associated with occupational radiation exposure dose. The finding showed body mass index (BMI) of radiology personnel had statistically significant association with occupational radiation exposure dose. The increase in body mass index of radiology personnel increases the occupational radiation exposure dose. The result is consistent with the study done in Addis Ababa, Ethiopia by Alemu Tesfahun [22]. The high scatter radiation received by radiology personnel with a high BMI might be one of the possible reasons. Since radiation protection equipment is created in free size, it may be shortened and washed for obese radiology personnel (radiology personnel with a high BMI). As a result, individuals might be able to carry out radiographic procedures without the use of radiation protection equipment. This might result in a higher occupational radiation dose for radiology personnel with a high BMI.

Radiology personnel's practice of timing had also statistically significant association with occupational radiation exposure dose. The radiology personnel's good practice of timing decreases the occupational radiation exposure dose. In other words, radiology personnel using short exposure time and wait some minute in exposure room after exposure, which is good practice of timing, had low occupational radiation exposure dose. This result is consistent with other studies done in Ethiopia [22] and other countries like Zambia [16], Malawi [30], India [35], Japan [36], and China [31]. The possible explanations might be related to effective practice of timing, which could improve the quality of primary radiation, limit the production of scattered radiation, and reduce the exposure of radiology personnel to scattered radiation.

Radiology personnel's working experience was statistically associated with occupational radiation exposure dose. As the working experience of radiology personnel increases, the occupational radiation exposure dose decreases. This result is consistent with the findings of two separate studies conducted in Iran [37, 38]. This might be related to the fact that as radiology professionals gain experience, their awareness of the biological effects of radiation increases, their practice of radiation protection improves, they develop a responsible attitude, the quality of their work improves, and the number of repeat x-rays decreases.

Radiology personnel's practice of distancing was statistically associated with mean annual deep dose equivalent. As radiology personnel's practice of distancing increases, mean annual deep dose equivalent decreases. This result is consistent with the study done in Zambia [16]. The possible reasons for this might be, reduction in the production and exposure of scattered radiation.

This study implies that increasing radiology personnel's experience through on-job training and experience sharing, improving radiology personnel's healthy lifestyle and improving radiology personnel's skill towards practice of timing and practice of distancing is mandatory.

The small size of the source population didn't allow using the intended sample size and in turn, limits the generalizability of the findings to similar settings in the country. In addition, the cross-sectional nature of the study did not enable us to establish strong causal relationships between the independent variables and occupational radiation exposure dose.

## Conclusions

This study showed that the annual shallow and deep dose equivalents were two times higher than the total medical exposure of world population. In addition to this, female radiology personnel have higher occupational radiation dose which is above the recommended fetal dose

level. The collective annual shallow and deep dose equivalents were also high. Although this study revealed that the mean annual shallow and deep dose equivalents were below the ERPA and ICRP, 2007 recommendations, the doses were higher compared to the studies in Ethiopia and other countries. Body mass index of radiology personnel, practice of timing, working experience, and practice of distancing were variables that had achieved statistically significant association for occupational radiation exposure dose. Therefore, improving radiology personnel's healthy lifestyle, developing radiology personnel's skills in practice of timing and practice of distancing, and increasing radiology personnel's experience are vital.

## Supporting information

**S1 Dataset.**
(DTA)

## Acknowledgments

We are grateful to the School of Public health, College of Medicine and Health Sciences, Wollo University and study participants for their valuable contribution during the study. Our acknowledgment is also extended to Amhara national regional state health bureau and voluntary health institutions and their radiology personnel for their contribution. Finally, we would like to thank Ethiopian radiation protection authority and their staff members, especially, Ato Sirur Kedir, director for notification and authorization directorate, and Ato Getnet Workneh and DosiMed testing laboratory, especially Ato Asgedom T/Michael, managing director of DosiMed laboratory and W/rt Rahel Teklay, laboratory technician.

## Author Contributions

**Conceptualization:** Tadesse Guangul Alemayehu, Getaw Walle Bazie.

**Data curation:** Tadesse Guangul Alemayehu, Getaw Walle Bazie.

**Formal analysis:** Tadesse Guangul Alemayehu, Getaw Walle Bazie.

**Investigation:** Tadesse Guangul Alemayehu, Getahun Gebre Bogale.

**Methodology:** Tadesse Guangul Alemayehu, Getahun Gebre Bogale, Getaw Walle Bazie.

**Project administration:** Tadesse Guangul Alemayehu, Getahun Gebre Bogale.

**Resources:** Getahun Gebre Bogale.

**Software:** Getahun Gebre Bogale, Getaw Walle Bazie.

**Supervision:** Tadesse Guangul Alemayehu, Getaw Walle Bazie.

**Writing – original draft:** Tadesse Guangul Alemayehu, Getahun Gebre Bogale, Getaw Walle Bazie.

**Writing – review & editing:** Tadesse Guangul Alemayehu, Getahun Gebre Bogale, Getaw Walle Bazie.

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
