## [Decision Letter · Decision Letter 0]

11 Oct 2022

PONE-D-22-24607Occupational Radiation Exposure Dose and Associated Factors among Radiology Personnel in Eastern Amhara, EthiopiaPLOS ONE

Dear Dr. Bazie,

Thank you for submitting your manuscript to PLOS ONE. After careful consideration, we feel that it has merit but does not fully meet PLOS ONE’s publication criteria as it currently stands. Therefore, we invite you to submit a revised version of the manuscript that addresses the points raised during the review process.This paper is about occupational exposure due to medical uses of radiation in eastern Amhara, Ethiopia. I am aware of this good scientific manuscript the authors produce. However both reviewers were highly critical of this manuscript and their opinions appear to be well-founded and convincingly elaborated. According to the PLOS ONE’s publication criteria, a scientific rationale for this work should be provided. What are missing in this study, are clearly references, discussions of the existing literature such as UNSCEAR 2020/2021 report, appropriate dose definitions and  more detail study design. In the interests of avoiding any signs of favoritism, I consider that the authors should re-submit after a careful major revision. Reviewer comments are given below.  

We look forward to receiving your revised manuscript.

Kind regards,

Sakae Kinase, Ph.D.

Academic Editor

PLOS ONE

Reviewers' comments:

Reviewer's Responses to Questions

**Comments to the Author**

1. Is the manuscript technically sound, and do the data support the conclusions?

Reviewer #1: Partly

Reviewer #2: Partly

2. Has the statistical analysis been performed appropriately and rigorously? 

Reviewer #1: I Don't Know

Reviewer #2: Yes

3. Have the authors made all data underlying the findings in their manuscript fully available?

Reviewer #1: Yes

Reviewer #2: Yes

4. Is the manuscript presented in an intelligible fashion and written in standard English?

Reviewer #1: Yes

Reviewer #2: Yes

5. Review Comments to the Author

Reviewer #1: Line 37 (Abstract): …those with a p-value < 0.5 were judged significant.

I thought you wanted to write “<0.05” instead of “<0.5”

Lines 47-48: “effective medical exposure of the world population”

It would be better to use technical terms such as effective dose correctly. I think you mean “global average of annual per caput effective dose due to medical exposure”.

Lines 159-161: The calculated mean (± SD) annual TLD readings…as occupational radiation exposure dose.

Line 186: Document review was used to extract the annual Thermoluminescent Dosimeter (TLD) readings…

I think short descriptions related to quality assurance of TLD measurement should be added. For example, TLD calibration procedures, types of TLD reading devices, and possibility of improper use of TLD are important for the quality assurance, I think.

Line 167: Collective dose: the sum of all individual effective doses

How to calculate “individual effective doses” should be described. I think these were calculated from “deep dose” and “shallow dose”. When protective equipment such as lead apron is used, attention should be paid for calculating effective doses from measured doses, as described in the following references.

Evaluation of the effectiveness of X-ray protective aprons in experimental and practical fields. Radiol Phys Technol (2014) 7:158–166, DOI 10.1007/s12194-013-0246-x

Effective dose to staff from interventional procedures: estimations from single and double dosimetry. Radiat Prot Dosim (2009), Vol. 136, No. 2, pp. 95–100

doi:10.1093/rpd/ncp155

Line 172: Ref [29] and Line 174: Ref [43]

I think these ref Nos. are wrong. Reference citation before Line 172 can be found in Line 139, which referred to No. [23].

Line 252

According to the text, lead apron, eye goggle, thyroid collar, and lead glove were used in some cases. In the case of using the lead apron, was TLD worn under the apron? If TLD was worn over the apron, shielding effects should be considered in estimating deep dose, shallow dose and effective dose, I think.

Lines 270-272, 275-286, 291, 292, 350-352, 364 and 368: effective deep and shallow doses

What are the definitions of “effective deep and shallow doses”? Are these different from “deep dose” and “shallow dose”? It should be clarified.

Lines 295-296

Deep and shallow doses were compared with effective dose (0.6 mSv). I think the definitions of these doses are different from each other. In particular, these dose values could be different even in the same exposure condition, when protective equipment was used (see comments on Line 167).

Also, studies from different countries were compared in Lines 303-312. It should be confirmed if these comparisons were done in terms of the same type of dose (e.g., effective dose).

Table 3 (lines 262-265)

More explanation is needed for items in “Leakage measurement” and “Collimator and beam alignment”, I think. In the text, explanation on Table 3 is too short (lines 262-265).

For example, the maximum dose for “work area dose result, door to the toilet” was 28 mSv. Is this within acceptable range?

Table 4

It would be better to consider significant figures. I think showing figures down to the second decimal would be enough.

Reviewer #2: The paper presents a study of occupational exposure due to medical applications in Ethopia. The paper is written in a clear and concise language. However the paper should be revised before publication. The study rationale and design should be explained in more detail and the implications of the study should be summarized. (e.g. Are there any measures taken to reduce doses/raise awareness for use of RPE?)

Some comments are given in the attachement.

6. PLOS authors have the option to publish the peer review history of their article (what does this mean?). If published, this will include your full peer review and any attached files.

Reviewer #1: No

Reviewer #2: No

---

## [Author Response · Author response to Decision Letter 0]

4 Apr 2023

Response to academic editor

Firstly, we would like to thank the academic editor for facilitating the process. We authors tried to address all your concerns such as the study rationale, references, appropriate dose definitions, and description of the study design. Please see the responses given for reviewers and the revised manuscript. 

Response to reviewers’ comments

We are grateful to reviewers for your valuable comments and questions that are helpful for improving the quality of the manuscript. The responses for each point are done as follows. 

Reviewer #1: 

• Line 37 (Abstract): …those with a p-value < 0.5 were judged significant.

I thought you wanted to write “<0.05” instead of “<0.5”

Response: Yes, thank you. That’s editorial error. (Please see the revised version). 

• Lines 47-48: “effective medical exposure of the world population”

It would be better to use technical terms such as effective dose correctly. I think you mean “global average of annual per caput effective dose due to medical exposure”.

Response: Yes, thank you. It is corrected accordingly.

• Lines 159-161: The calculated mean (± SD) annual TLD readings…as occupational radiation exposure dose.

Response: The individual radiation doses were regularly read and recorded by ERPA and DosiMed testing laboratory, which are the only two organizations giving this service in Ethiopia currently. By document review (secondary data) the annual doses of radiology personnel (TLD readings) were extracted from the databases of ERPA and DosiMed testing laboratory. Then, mean (± SD) were calculated.

• Line 186: Document review was used to extract the annual Thermoluminescent Dosimeter (TLD) readings…

I think short descriptions related to quality assurance of TLD measurement should be added. For example, TLD calibration procedures, types of TLD reading devices, and possibility of improper use of TLD are important for the quality assurance, I think.

Response: The annual TLD readings were taken from the ERPA and DosiMed testing laboratory (secondary data) for this study. Computerized Vinten Solaro machine was used by the two organizations as a TLD reader. The quality assurance of each TLD machine was done regularly by the experts in the two organizations. Furthermore, each radiology personnel were equipped with two TLDs (when one is sent for reading, the other is used). The radiology personnel were responsible to declare the TLD they use every two months by the Authority. So, the possibilities of improper use of TLD were very unlikely. (Please see the revised version)

• Line 167: Collective dose: the sum of all individual effective doses

How to calculate “individual effective doses” should be described. I think these were calculated from “deep dose” and “shallow dose”. When protective equipment such as lead apron is used, attention should be paid for calculating effective doses from measured doses, as described in the following references.

Evaluation of the effectiveness of X-ray protective aprons in experimental and practical fields. Radiol Phys Technol (2014) 7:158–166, DOI 10.1007/s12194-013-0246-x

Effective dose to staff from interventional procedures: estimations from single and double dosimetry. Radiat Prot Dosim (2009), Vol. 136, No. 2, pp. 95–100

doi:10.1093/rpd/ncp155

Response: The individual effective dose was calculated from the deep dose (HP10) and shallow dose (HP 0.007), which indicates the personal dose at a tissue depth of 10 mm and 0.007 cm. Using the lead apron, does not affect the individual effective dose as the TLD is put under the apron. (Please see the revised version)

• Line 172: Ref [29] and Line 174: Ref [43]

I think these ref Nos. are wrong. Reference citation before Line 172 can be found in Line 139, which referred to No. [23].

Response: Thank you. Ref [29] is changed to ref [24] and Ref [43] is changed to Ref [25]. All the references after Ref [23] are corrected accordingly. 

• Line 252: According to the text, lead apron, eye goggle, thyroid collar, and lead glove were used in some cases. In the case of using the lead apron, was TLD worn under the apron? If TLD was worn over the apron, shielding effects should be considered in estimating deep dose, shallow dose and effective dose, I think.

Response: Thank you, it is a great concern. The TLD were worn under the lead apron. This is highly recommended by ERPA, the governmental authority in Ethiopia, and the radiology personnel were aware of that. 

• Lines 270-272, 275-286, 291, 292, 350-352, 364 and 368: effective deep and shallow doses.

What are the definitions of “effective deep and shallow doses”? Are these different from “deep dose” and “shallow dose”? It should be clarified.

Response: in case of TLD measurements and for our study, no difference. Effective deep dose or simply deep dose is a whole-body dose. Shallow dose or effective shallow dose (a shallow dose equivalent to the whole body) is a skin dose.

• Lines 295-296: Deep and shallow doses were compared with effective dose (0.6 mSv). I think the definitions of these doses are different from each other. In particular, these dose values could be different even in the same exposure condition, when protective equipment was used (see comments on Line 167).

Also, studies from different countries were compared in Lines 303-312. It should be confirmed if these comparisons were done in terms of the same type of dose (e.g., effective dose).

Response: It is to mean the annual effective deep and shallow dose. It is corrected in the revised version.

• Table 3 (lines 262-265): More explanation is needed for items in “Leakage measurement” and “Collimator and beam alignment”, I think. In the text, explanation on Table 3 is too short (lines 262-265).

For example, the maximum dose for “work area dose result, door to the toilet” was 28 mSv. Is this within acceptable range?

Response: 

All the quality control test results were taken from ERPA data base by document review, which was done by the experts of ERPA. Leakage measurement {a measure of leakage radiation from the radiation source assembly (secondary radiation)} was measured at the collimator and at the tube. It is deviated acceptably by 0.24% both at collimator and at the tube. 

Collimator and beam alignment: the beam alignment was deviated from the long axis of the collimator (long axis variance) from -1.25% to +1.5% and from the short axis of the collimator (short axis variance) from -1.25 to +1.25.

Work area dose result: the measurement of radiation doses of areas adjacent to the x-ray room. It is 0.28mSv. It is writing error. The dose result from the door to the toilet was from 0.09 to 0.28 (sorry! It was editorial error and still it was deviated in acceptable range) (Please see the revised table 3).

• Table 4: It would be better to consider significant figures. I think showing figures down to the second decimal would be enough.

Response: Okay, it is corrected accordingly. (Please see the revised table 4).

Reviewer #2: 

The paper presents a study of occupational exposure due to medical applications in Ethiopia. The paper is written in a clear and concise language. However, the paper should be revised before publication. The study rationale and design should be explained in more detail and the implications of the study should be summarized. (E.g., are there any measures taken to reduce doses/raise awareness for use of RPE?)

Response: Thank you dear reviewer. The study rationale is explained from lines 100-117 of the revised track changed manuscript. The study design is cross-sectional study, i.e., both occupational radiation exposure dose and its associated factors are assessed at a point in time. Yes of course, training was given for radiology personnel about radiation effect, radiation protection, and proper use of RPE.

---

## [Editor Report · Decision Letter 1]

6 Apr 2023

PONE-D-22-24607R1Occupational Radiation Exposure Dose and Associated Factors among Radiology Personnel in Eastern Amhara, EthiopiaPLOS ONE

Dear Dr. Bazie,

Thank you for submitting your manuscript to PLOS ONE. After careful consideration, we feel that it has merit but does not fully meet PLOS ONE’s publication criteria as it currently stands. Therefore, we invite you to submit a revised version of the manuscript that addresses the points raised during the review process.

We look forward to receiving your revised manuscript.

Kind regards,

Sakae Kinase, Ph.D.

Academic Editor

PLOS ONE

**Additional Editor Comments:**

The manuscript indicates that the authors should improve their knowledge of basic principles of dosimetry and dose assessment, specifically, in relation to concepts of external doses. In the US, deep dose equivalent and shallow dose equivalent are used as operational quantities. Personal dose equivalent, Hp(10) is called the deep dose equivalent. Personal dose equivalent, Hp(0.07) is called the shallow dose equivalent. However, the authors should use appropriate terminology for protection/operational quantities defined by ICRP/ICRU. Recommendations of ICRP and the UNSCEAR reports can be found very useful for this. In addition, the authors should improve the manuscript according to reviewer's comments. Current version has been with changes highlighted point by point according to reviewer 1's comments. Please revise and improve the manuscript according to reviewer 2's comments.

Reviewer 2's comments are as follows:

page 2

line 37 p<0.05 were judged significant (a 0 is missing here)

line 42 and 44 some values are printed in bold, this should be removed

page 3

line 47 "effective medical exposures of the world population" - What is meant by this? Please give a value and a reference for the definition

line 60 ultrasound and MRI are examples for use/application of non ionizing radiation. Please check and reword the sentence.

line 68 "it is preferable no to ..." Is this really a preference? I guess it is a requirement/regulation

Page 6

line 132 DosiMed should be explained

line 139 please add an explanation for marginal error

page 7

line 162 what does regularly mean here? Annual? Monthly? Daily)

page 8

line 182 "different literatures" - please provide references

page 9

line 205 Cronbach's alpha - definition and reference should be provided here

page 10

line 229 declaration of Helsinki, a reference should be provided.

page 11

line 246/table 1 - Religious believe, marital status and income should be removed as these are not required for the understanding of the study and are not evaluated further.

line 249 birr - a conversion to US Dollars should be provided so that international readers could comply this. I suggest to remove this information on income as well.

page 12

line 258 The listing of the machines should be extended. In my understanding 57 institution s used 60 machines? Has a correlation been checked for type of machine and dose?

page 13

lines 290-293 This paragraph should be removed as it repeats the information from the previous section.

page 14

line 301 What is meant by "government's concern to radiation"? Please explain.

line 303 A Table compiling the results of the studies from different countries would be helpful here. Consider adding one here.

page 15

line 329-331 What is meant by "free size"? Please check the sentence and reword more clearly. Dou you want to say that the RPE might be to small/not fitting properly and thus the probability for not being used is lager for higher BMI?

page 17

line 376 - abbreviations should be given in the text at the first occurrence, see e.g. line 60: "magnetic resonance imaging (MRI)"

page 18

format of the references should be checked, especially 3,9,11,14,17

---

## [Author Response · Author response to Decision Letter 1]

19 Apr 2023

Response to academic editor

Additional Editor Comments:

The manuscript indicates that the authors should improve their knowledge of basic principles of dosimetry and dose assessment, specifically, in relation to concepts of external doses. In the US, deep dose equivalent and shallow dose equivalent are used as operational quantities. Personal dose equivalent, Hp (10) is called the deep dose equivalent. Personal dose equivalent, Hp (0.07) is called the shallow dose equivalent. However, the authors should use appropriate terminology for protection/operational quantities defined by ICRP/ICRU. Recommendations of ICRP and the UNSCEAR reports can be found very useful for this. In addition, the authors should improve the manuscript according to reviewer's comments. Current version has been with changes highlighted point by point according to reviewer 1's comments. Please revise and improve the manuscript according to reviewer 2's comments.

Response: Thank you so much. All your concerns such as clarification of terms, recommendations of ICRP and UNSCEAR reports, and comments of both the reviewers are considered. The manuscript is revised considering both the academic editor and reviewers’ comments. 

Response to reviewer 1 comments

We are grateful to reviewer 1 for your valuable comments and questions that are helpful for improving the quality of the manuscript. The responses for each point are done as follows. 

Reviewer #1: 

• Line 37 (Abstract): …those with a p-value < 0.5 were judged significant.

I thought you wanted to write “<0.05” instead of “<0.5”

Response: Yes, thank you. That’s editorial error. (Please see the revised version). 

• Lines 47-48: “effective medical exposure of the world population”

It would be better to use technical terms such as effective dose correctly. I think you mean “global average of annual per caput effective dose due to medical exposure”.

Response: Yes, thank you. It is corrected accordingly.

• Lines 159-161: The calculated mean (± SD) annual TLD readings…as occupational radiation exposure dose.

Response: The individual radiation doses were regularly read and recorded by ERPA and DosiMed testing laboratory, which are the only two organizations giving this service in Ethiopia currently. By document review (secondary data) the annual doses of radiology personnel (TLD readings) were extracted from the databases of ERPA and DosiMed testing laboratory. Then, mean (± SD) were calculated.

• Line 186: Document review was used to extract the annual Thermoluminescent Dosimeter (TLD) readings…

I think short descriptions related to quality assurance of TLD measurement should be added. For example, TLD calibration procedures, types of TLD reading devices, and possibility of improper use of TLD are important for the quality assurance, I think.

Response: The annual TLD readings were taken from the ERPA and DosiMed testing laboratory (secondary data) for this study. Computerized Vinten Solaro machine was used by the two organizations as a TLD reader. The quality assurance of each TLD machine was done regularly by the experts in the two organizations. Furthermore, each radiology personnel were equipped with two TLDs (when one is sent for reading, the other is used). The radiology personnel were responsible to declare the TLD they use every two months by the Authority. So, the possibilities of improper use of TLD were very unlikely. (Please see the revised version)

• Line 167: Collective dose: the sum of all individual effective doses

How to calculate “individual effective doses” should be described. I think these were calculated from “deep dose” and “shallow dose”. When protective equipment such as lead apron is used, attention should be paid for calculating effective doses from measured doses, as described in the following references.

Evaluation of the effectiveness of X-ray protective aprons in experimental and practical fields. Radiol Phys Technol (2014) 7:158–166, DOI 10.1007/s12194-013-0246-x

Effective dose to staff from interventional procedures: estimations from single and double dosimetry. Radiat Prot Dosim (2009), Vol. 136, No. 2, pp. 95–100

doi:10.1093/rpd/ncp155

Response: The individual effective dose was calculated from the deep dose (HP10) and shallow dose (HP 0.007), which indicates the personal dose at a tissue depth of 10 mm and 0.007 cm. Using the lead apron, does not affect the individual effective dose as the TLD is put under the apron. (Please see the revised version)

• Line 172: Ref [29] and Line 174: Ref [43] I think these ref Nos. are wrong. Reference citation before Line 172 can be found in Line 139, which referred to No. [23].

Response: Thank you. Ref [29] is changed to ref [24] and Ref [43] is changed to Ref [25]. All the references after Ref [23] are corrected accordingly. 

• Line 252: According to the text, lead apron, eye goggle, thyroid collar, and lead glove were used in some cases. In the case of using the lead apron, was TLD worn under the apron? If TLD was worn over the apron, shielding effects should be considered in estimating deep dose, shallow dose and effective dose, I think.

Response: Thank you, it is a great concern. The TLD were worn under the lead apron. This is highly recommended by ERPA, the governmental authority in Ethiopia, and the radiology personnel were aware of that. 

• Lines 270-272, 275-286, 291, 292, 350-352, 364 and 368: effective deep and shallow doses.

What are the definitions of “effective deep and shallow doses”? Are these different from “deep dose” and “shallow dose”? It should be clarified.

Response: in case of TLD measurements and for our study, no difference. Effective deep dose or simply deep dose is a whole-body dose. Shallow dose or effective shallow dose (a shallow dose equivalent to the whole body) is a skin dose.

• Lines 295-296: Deep and shallow doses were compared with effective dose (0.6 mSv). I think the definitions of these doses are different from each other. In particular, these dose values could be different even in the same exposure condition, when protective equipment was used (see comments on Line 167).

Also, studies from different countries were compared in Lines 303-312. It should be confirmed if these comparisons were done in terms of the same type of dose (e.g., effective dose).

Response: It is to mean the annual effective deep and shallow dose. It is corrected in the revised version.

• Table 3 (lines 262-265): More explanation is needed for items in “Leakage measurement” and “Collimator and beam alignment”, I think. In the text, explanation on Table 3 is too short (lines 262-265).

For example, the maximum dose for “work area dose result, door to the toilet” was 28 mSv. Is this within acceptable range?

Response: 

All the quality control test results were taken from ERPA data base by document review, which was done by the experts of ERPA. Leakage measurement {a measure of leakage radiation from the radiation source assembly (secondary radiation)} was measured at the collimator and at the tube. It is deviated acceptably by 0.24% both at collimator and at the tube. 

Collimator and beam alignment: the beam alignment was deviated from the long axis of the collimator (long axis variance) from -1.25% to +1.5% and from the short axis of the collimator (short axis variance) from -1.25 to +1.25.

Work area dose result: the measurement of radiation doses of areas adjacent to the x-ray room. It is 0.28mSv. It is writing error. The dose result from the door to the toilet was from 0.09 to 0.28 (sorry! It was editorial error and still it was deviated in acceptable range) (Please see the revised table 3).

• Table 4: It would be better to consider significant figures. I think showing figures down to the second decimal would be enough.

Response: Okay, it is corrected accordingly. (Please see the revised table 4).

Response to reviewer 2 comments

We are grateful to reviewer 2 for the valuable comments and questions that are helpful for improving the quality of the manuscript. The responses for each point are done as follows: 

page 2

line 37 p<0.05 were judged significant (a 0 is missing here)

Response: Yes, thank you. That’s editorial error. (Please see the revised version).

line 42 and 44 some values are printed in bold, this should be removed

Response: Ok! Thank you, it is corrected accordingly.

page 3

line 47 "effective medical exposures of the world population" - What is meant by this? Please give a value and a reference for the definition

Response: it is to mean “global average of annual per caput effective dose due to medical exposure”. It is 0.6 mSv. Reference no. 9 in the mother document (UNSCEAR 2010 report)

line 60 ultrasound and MRI are examples for use/application of non-ionizing radiation. Please check and reword the sentence.

Response: Non-ionizing radiation is a form of radiation with less energy than ionizing radiation. Examples: high frequency sound waves (used in ultrasound) and radiofrequency (used in MRI). Rewording is done in the revised manuscript. 

line 68 "it is preferable no to ..." Is this really a preference? I guess it is a requirement/regulation

Response: The yearly workers’ effective dose limit is 12 mSv per year which is the requirement/regulation for both male and female workers. 1 mSv per year is the recommended yearly fetal dosage limit. Hence, for female workers it is a preference not to exceed this dosage limit which is thought to be potential of pregnancy. However, for those known pregnant workers 1 mSv per year is a regulation and the preference is not to be exposed to radiation at all.

Page 6

line 132 DosiMed should be explained

Response: The DosiMed testing laboratory is a private company in Ethiopia providing personal dosimeters for workers in different medical and industrial facilities from both government and private sectors.

line 139 please add an explanation for marginal error

Response: The margin of error is also called maximum tolerable error is an error assumed to be committed by researchers when calculating the minimum sample size required for estimating the outcome variable (i.e., occupational radiation exposure dose in the current study). In the current study, 5% is used that is the maximum percentage assumed. 

page 7

line 162 what does regularly mean here? Annual? Monthly? Daily)

Response: Every two months. It is corrected accordingly. 

page 8

line 182 "different literatures" - please provide references

Response: The references are used are 10, 15, 16, 22, 29, 33 & 37. They are indicated in the revised manuscript. 

page 9

line 205 Cronbach's alpha - definition and reference should be provided here

Response: Cronbach's alpha is a measure of internal consistency, that is, how closely related a set of items are as a group. Cronbach's alpha reliability coefficient normally ranges between 0 and 1. If the Cronbach’s alpha value is greater than or equal to 0.7, it is considered as the tool is reliable. Modifications are done in the revised manuscript accordingly. 

page 10

line 229 declaration of Helsinki, a reference should be provided.

Response: The sentence is removed from the revised version. 

page 11

line 246/table 1 - Religious believe, marital status and income should be removed as these are not required for the understanding of the study and are not evaluated further.

Response: They are included just to know the frequency and percentage, not for further analysis. 

line 249 birr - a conversion to US Dollars should be provided so that international readers could comply this. I suggest to remove this information on income as well.

Response: Information regarding income is removed from the revised manuscript. 

page 12

line 258 The listing of the machines should be extended. In my understanding 57 institution s used 60 machines? Has a correlation been checked for type of machine and dose?

Response: Yes of course, 60 machines were used by 57 health institutions. This is because there were health institutions using more than one type of machine (one government hospital used x-ray and CT scan machine, 2 private health institutions used x-ray and fluoroscopy machines and 1 diagnostic center used CT scan machine only. The correlation was checked.

page 13

lines 290-293 This paragraph should be removed as it repeats the information from the previous section.

Response: It is put to show the principal findings in the very beginning of the discussion section. 

page 14

line 301 What is meant by "government's concern to radiation"? Please explain.

Response: "government's concern to radiation”: - it is to mean somehow there is a gap in monitoring of the regulations in Ethiopia.

line 303 A Table compiling the results of the studies from different countries would be helpful here. Consider adding one here.

Response: It is added in the revised manuscript accordingly. 

Table 6: The mean annual effective deep dose of selected countries in the world, 2023

Countries Radiation dose in mSv remark

Ghana 1.05 

Kuwait 1.03 

Kenya 2.15 

Malawi 2.964 

China 0.35 

Ireland 0.32 

Saudi Arabia 0.41 

UAE 0.39-0.83 

Pakistan 0.29-0.80 

Ethiopia 0.41 Previous study

Ethiopia 1.2 (Shallow dose) and 1.02 (Deep dose) Current study 

page 15

line 329-331 What is meant by "free size"? Please check the sentence and reword more clearly. Dou you want to say that the RPE might be too small/not fitting properly and thus the probability for not being used is lager for higher BMI?

Response: yes, absolutely. The RPE is prepared in average size. So, it might be shortened and washed (not fitted properly) for those with high BMI.

page 17

line 376 - abbreviations should be given in the text at the first occurrence, see e.g., line 60: "magnetic resonance imaging (MRI)"

Response: The revised manuscript is done that way. 

page 18

format of the references should be checked, especially 3,9,11,14,17

Response: Corrected accordingly. Please see the revised manuscript.

---

## [Decision Letter · Decision Letter 2]

2 May 2023

PONE-D-22-24607R2Occupational Radiation Exposure Dose and Associated Factors among Radiology Personnel in Eastern Amhara, EthiopiaPLOS ONE

Dear Dr. Bazie,

Thank you for submitting your manuscript to PLOS ONE. After careful consideration, we feel that it has merit but does not fully meet PLOS ONE’s publication criteria as it currently stands. Therefore, we invite you to submit a revised version of the manuscript that addresses the points raised during the review process.

We look forward to receiving your revised manuscript.

Kind regards,

Sakae Kinase, Ph.D.

Academic Editor

PLOS ONE

Journal Requirements:

Reviewers' comments:

Reviewer's Responses to Questions

**Comments to the Author**

1. If the authors have adequately addressed your comments raised in a previous round of review and you feel that this manuscript is now acceptable for publication, you may indicate that here to bypass the “Comments to the Author” section, enter your conflict of interest statement in the “Confidential to Editor” section, and submit your "Accept" recommendation.

Reviewer #1: (No Response)

Reviewer #2: All comments have been addressed

2. Is the manuscript technically sound, and do the data support the conclusions?

Reviewer #1: Partly

Reviewer #2: Yes

3. Has the statistical analysis been performed appropriately and rigorously? 

Reviewer #1: Yes

Reviewer #2: Yes

4. Have the authors made all data underlying the findings in their manuscript fully available?

Reviewer #1: Yes

Reviewer #2: Yes

5. Is the manuscript presented in an intelligible fashion and written in standard English?

Reviewer #1: Yes

Reviewer #2: Yes

6. Review Comments to the Author

Reviewer #1: Almost all my previous comments seem to be considered by the authors. I appreciate the authors’ efforts for revising the manuscript. The only one point that I’d like to address again is the use of terminology. It is also commented from the academic editor.

According to your response shown below, there is no difference between “effective deep (or shallow) dose” and “deep (or shallow) dose”. If there is no difference between the two technical terms, I would suggest using “deep dose” and “shallow dose” to avoid confusion.

My previous comments

Lines 270-272, 275-286, 291, 292, 350-352, 364 and 368: effective deep and shallow doses

What are the definitions of “effective deep and shallow doses”? Are these different from “deep dose” and “shallow dose”? It should be clarified.

Your response

In case of TLD measurements and for our study, no difference. Effective deep dose or simply deep dose is a whole-body dose. Shallow dose or effective shallow dose (a shallow dose equivalent to the whole body) is a skin dose.

Reviewer #2: All comments have been properly addressed and the text is improved and in my eyes ready for publicaiton.

7. PLOS authors have the option to publish the peer review history of their article (what does this mean?). If published, this will include your full peer review and any attached files.

Reviewer #1: No

Reviewer #2: No

---

## [Author Response · Author response to Decision Letter 2]

4 May 2023

Response to Academic Editor

Response: Thank you so much. We tried to revise the references as per the journal requirement. We didn’t get retracted articles. Reference 2 is changed by the recent reference. Please see the track changed manuscript to see the specific changes made on the references list. Finally, the manuscript is revised considering both the academic editor and reviewer #1 comments. 

Response to Reviewer 1 Comments

Reviewer #1: Almost all my previous comments seem to be considered by the authors. I appreciate the authors’ efforts for revising the manuscript. The only one point that I’d like to address again is the use of terminology. It is also commented from the academic editor.

According to your response shown below, there is no difference between “effective deep (or shallow) dose” and “deep (or shallow) dose”. If there is no difference between the two technical terms, I would suggest using “deep dose” and “shallow dose” to avoid confusion.

My previous comments

Lines 270-272, 275-286, 291, 292, 350-352, 364 and 368: effective deep and shallow doses

What are the definitions of “effective deep and shallow doses”? Are these different from “deep dose” and “shallow dose”? It should be clarified.

Your response

In case of TLD measurements and for our study, no difference. Effective deep dose or simply deep dose is a whole-body dose. Shallow dose or effective shallow dose (a shallow dose equivalent to the whole body) is a skin dose.

Response: Dear reviewer #1, thank you so much for the time you invested on the manuscript and the scientific ideas you suggested us to make the manuscript improved. 

Regarding the terminologies “effective deep (or shallow) dose” and “deep (or shallow) dose”, in our study, since it is done on the whole body (not on a specific organ), deep dose and effective deep dose are the same. Similarly, shallow dose and effective shallow dose are the same. Therefore, as per your suggestion, the manuscript is corrected. Deep and shallow dose are used instead of effective deep and shallow dose.

---

## [Editor Report · Decision Letter 3]

12 May 2023

PONE-D-22-24607R3Occupational Radiation Exposure Dose and Associated Factors among Radiology Personnel in Eastern Amhara, EthiopiaPLOS ONE

Dear Dr. Bazie,

Thank you for submitting your manuscript to PLOS ONE. After careful consideration, we feel that it has merit but does not fully meet PLOS ONE’s publication criteria as it currently stands. Therefore, we invite you to submit a revised version of the manuscript that addresses the points raised during the review process.

We look forward to receiving your revised manuscript.

Kind regards,

Sakae Kinase, Ph.D.

Academic Editor

PLOS ONE

Journal Requirements:

Additional Editor Comments:

This paper is much better in its present version. However, I still have some proposals for further improvements to make: Please use appropriate terminology on dose concept. The authors should check the definitions of the deep/shallow dose equivalent. Personal dose equivalent, Hp(10) and Hp(0.07) is an operational quantity for individual monitoring. Hp(10) is called the deep dose equivalent. Hp(0.07) is called the shallow dose equivalent. In addition, person-Sv is used for collective effective dose.

---

## [Author Response · Author response to Decision Letter 3]

13 May 2023

Response to Academic Editor

Journal Requirements:

Response: The reference list is complete and correct and we didn’t get retracted articles. 

Additional Editor Comments:

This paper is much better in its present version. However, I still have some proposals for further improvements to make: Please use appropriate terminology on dose concept. The authors should check the definitions of the deep/shallow dose equivalent. Personal dose equivalent, Hp(10) and Hp(0.07) is an operational quantity for individual monitoring. Hp(10) is called the deep dose equivalent. Hp(0.07) is called the shallow dose equivalent. In addition, person-Sv is used for collective effective dose.

Response: Thank you so much for your concern. The terms deep dose and shallow dose are changed by deep dose equivalent and shallow dose equivalent. The definitions are also checked and it is correct. We used person-Sv instead of man-Sv for collective effective dose.

---

## [Editor Report · Decision Letter 4]

16 May 2023

Occupational Radiation Exposure Dose and Associated Factors among Radiology Personnel in Eastern Amhara, Ethiopia

PONE-D-22-24607R4

Dear Dr. Bazie,

We’re pleased to inform you that your manuscript has been judged scientifically suitable for publication and will be formally accepted for publication once it meets all outstanding technical requirements.

Kind regards,

Sakae Kinase, Ph.D.

Academic Editor

PLOS ONE

Additional Editor Comments (optional):

I am pleased to inform you that your revised manuscript has been accepted for publication in PLOS ONE as a Research Article. Page 8 line 169 Hp(10) and Hp(0.07) are normally used in radiation protection dosimetry. Page 8 line 170, 0.007 cm 0.07mm. You have already mentioned 0.07 mm in line 165.
---

## [Editor Report · Acceptance letter]

18 May 2023

PONE-D-22-24607R4 

Occupational Radiation Exposure Dose and Associated Factors among Radiology Personnel in Eastern Amhara, Ethiopia 

Dear Dr. Bazie:

I'm pleased to inform you that your manuscript has been deemed suitable for publication in PLOS ONE. Congratulations! Your manuscript is now with our production department. 

Kind regards, 

on behalf of

Professor Sakae Kinase 

Academic Editor

PLOS ONE